# hdl2v: A Code Translation Dataset for Enhanced LLM Verilog Generation

Charles Hong*, Brendan Roberts, Huijae An, Alex Um, Advay Ratan, Yakun Sophia Shao

UC Berkeley

*charleshong@berkeley.edu

*Abstract*—Large language models (LLMs) are playing an increasingly large role in domains such as code generation, including hardware code generation, where Verilog is the key language. However, the amount of publicly available Verilog code pales in comparison to the amount of code available for software languages like Python. In this work, we present hdl2v ("HDL-to-Verilog"), a dataset which seeks to increase the amount of available human-written Verilog data by translating or compiling three other hardware description languages—VHDL, Chisel, and PyMTL3—to Verilog. Furthermore, we demonstrate the value of hdl2v in enhancing LLM Verilog generation by improving performance of a 32 billion-parameter open-weight model by up to 23% (pass@10) in VerilogEvalV2, without utilizing any data augmentation or knowledge distillation from larger models. We also show hdl2v's ability to boost the performance of a data augmentation-based fine-tuning approach by 63%. Finally, we characterize and analyze our dataset to better understand which characteristics of HDL-to-Verilog datasets can be expanded upon in future work for even better performance.

## I. INTRODUCTION

Large language models (LLMs) have demonstrated impressive performance in a wide range of tasks, ranging from general reasoning ability and instruction-following [6], [25] to code generation [9], [18], [24], [27]. LLMs have potential to automate a wide range of tasks in hardware design, ranging from design to verification to optimization [5], [8], [14], [15]. A number of studies have attempted to evaluate and improve LLMs' potential in generating Verilog code [13], [31], [36].

Compared to popular software programming languages such as Python or C, there is not as much publicly available Verilog. In fact, as of April 2025, there are 132,264 GitHub repositories with Python as the primary language, compared to just 848 for Verilog or SystemVerilog [1]. As a result, a number of prior works have attempted to fine-tune LLMs on novel Verilog datasets, and successfully improved Verilog generation performance. These works utilize techniques such as data augmentation [7], [10], [28], [33], [35] and synthetic Verilog generation [21].

However, Verilog is not the only hardware description language (HDL). While Verilog is often used as the common interface between hardware code and tools such as RTL simulators or synthesis software, design can be done in higher-level languages such as Chisel. VHDL is another popular HDL with its own ecosystem of supported hardware. Nonetheless, generating Verilog with LLMs is still an important task as it is the most commonly supported and written.

In this work, we investigate how the wealth of HDL code in languages other than Verilog can be used to improve LLMs' ability to generate Verilog. Specifically, we present hdl2v, a dataset consisting of 46,549 pairs of VHDL, Chisel, and PyMTL3 translated/compiled to Verilog. We use this data in supervised fine-tuning of LLMs. Our findings from these experiments are as follows:

- Fine-tuning with this data yields significant improvements in Verilog generation performance. We find that VerilogEvalV2 performance of a state-of-the-art open-weight LLM improves by up to 13% for pass@1 and 23% for pass@10 upon being fine-tuned on a combination of our datasets.
- hdl2v works in tandem with other fine-tuning approaches. We demonstrate that by adding data from hdl2v to existing Verilog training data, we boost performance of a data augmentation approach by 63%.
- Language matters; fine-tuning with VHDL-Verilog pairs yields better results than C-Verilog pairs, when the Verilog is held constant.
- Fine-tuned models learn from the code in prompt-response pairs, not just natural language. However, utilizing meaningful module and variable names is important in helping LLMs learn from this data.

hdl2v is fully open-source and available for others to expand on this research. [1] [2] [3]

## II. BACKGROUND

### A. Verilog Code Generation

Prior work has sought to improve LLMs' ability to generate correct Verilog code, using techniques such as fine tuning on textbooks and Verilog from Github [31]. Other prior works augment existing Verilog datasets [7], [28], [33], [35] or generate novel synthetic Verilog [21]. Multi-agent systems utilize feedback from RTL simulation tools and modify test code in order to debug generated code [13], [36]. Benchmarks such as VerilogEval [20] and RTLLM [22] have been developed to standardize evaluation of LLM Verilog generation performance.

This work does not seek to supercede such prior work. Instead, we provide new data that can complement other

---

[1] VHDL dataset: https://huggingface.co/datasets/hdl2v/vhdl-dataset
[2] Chisel dataset: https://huggingface.co/datasets/hdl2v/chisel-dataset
[3] PyMTL3 dataset: https://huggingface.co/datasets/hdl2v/pymtl3-dataset

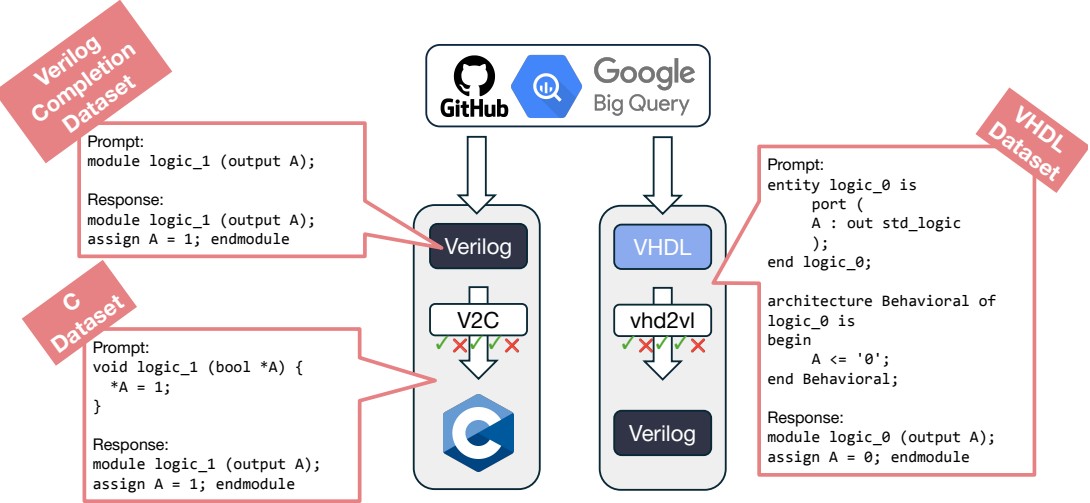

Fig. 1. How the Verilog Completion, C, and VHDL datasets are collected. Note that Verilog is translated to C and VHDL is translated to Verilog, but during fine-tuning Verilog is always used as the response.

approaches, such as data augmentation and agentic systems, to further improve LLM Verilog generation. We show a concrete example of this synergy in Section V-C.

### B. Hardware Description Languages (HDLs)

Verilog is a key language in hardware design due to its use as a common representation for representing register-transfer level (RTL) designs. A wide range of design automation tools, such as logic synthesis software and FPGA compilers, use Verilog as a common input to represent digital hardware. However, while digital hardware design is commonly done in Verilog (and its more feature-rich descendant, SystemVerilog), many other hardware description languages (HDLs) exist and have been used to tape out real-world chips. For example, VHDL is an alternative to Verilog that fills largely the same role.

On the other hand, high-level HDLs like Chisel [3], Spinal-HDL [26], MyHDL [11], PyMTL3 [19], and many others seek to provide user-friendly features such as type-checking and greater parameterizability. Often these languages are embedded in high-level software languages, such as Scala in the case of Chisel and SpinalHDL, and Python in the case of MyHDL and PyMTL3. Nonetheless, designs written in these languages still compile to Verilog so that they can be used with industry standard design automation tools. As a result, the Chisel or PyMTL3 code is often a higher-level (and as a result, closer to natural language) representation of the Verilog it compiles to. We use this fact to our advantage in hdl2v—specifically, we show that fine-tuning LLMs with pairs of Chisel/PyMTL3 code and their corresponding compiled Verilog can successfully improve LLMs' ability to generate Verilog from a natural language spec.

We select VHDL for this work as it is the most popular HDL outside of Verilog and SytemVerilog. Additionally, we include Chisel and PyMTL3 as they are two of the most popular high-level HDLs, embedded in two different high-level software languages, with a large amount of diverse and high-quality code written in each language. Of course, future work could extend our approach to include other HDLs.

### C. Verilog Translation for LLM Fine-Tuning

BetterV [28] introduces the idea of translating Verilog to C in order to improve correctness of generated Verilog. However, we demonstrate in Section V-A that when training with a Verilog dataset and the corresponding data translated to C, the benefit from training with C is minimal compared to training with only Verilog. Furthermore, the Verilog in this dataset is also likely to be present in LLMs' pre-training data, as it originates from open-source Github repositories.

To our knowledge, this work is the first to translate other HDLs to Verilog to generate novel Verilog data for LLM fine-tuning.

### III. DATASET CONSTRUCTION

### A. Datasets From Prior Work

As a baseline, we fine-tune with datasets from prior work that are based on existing Verilog from public sources.

*1) Verilog Completion:* As in BetterV [28], this dataset consists of a filtered set of Verilog modules from public sources. In this case, the prompt for fine-tuning is the header of the Verilog module, and the response is the rest of the module. Figure 1 shows an example of what an entry in this dataset might look like. This dataset contains 147,138 entries with total size 84.4 MB.

*2) C:* As in BetterV [28] and as depicted in Figure 1, we use v2c [23] to translate the above Verilog to C. In this case, the prompt for fine-tuning is actually the translated C code, and the original Verilog is the response. The intent is to improve the model's understanding of the Verilog code by attempting to correlate it with C code, which has greater presence in pretraining data. As not all Verilog modules in our dataset can

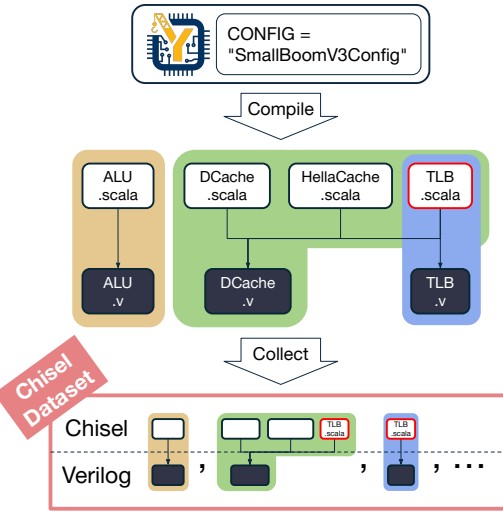

Fig. 2. How the Chisel dataset is collected. One pair is collected for each generated Verilog module. Note that in cases where multiple Verilog modules are generated from the same Chisel file, that Chisel file will be included in multiple pairs in the dataset.

be successfully translated to C, this results in 26,803 entries with size 36.5 MB.

### B. hdl2v Datasets

Each hdl2v dataset is fully open-source and available on HuggingFace, as described in Section I.

*1) VHDL:* As shown in Figure 1, we use Google BigQuery to collect VHDL files from Github. Specifically, we collect every file with a `.vhd` or `.vhdl` extension. This results in 53,698 VHDL entities. We attempt to translate each VHDL entity to Verilog using the open-source tool vhd2vl [12]. vhd2vl is able to successfully translate 8974 entities to Verilog, and we filter out entries that do not contain the strings `module` and later `endmodule`. The remaining 8626 entries have a total size of 48.4 MB. Each prompt is a VHDL entity, and its response is the corresponding translated Verilog module.

*2) Chisel:* Chisel is a high-level HDL embedded in Scala that can be compiled into Verilog or SystemVerilog. Therefore, it intrinsically provides matching pairs between itself and Verilog. To gather this data, we used Chipyard [2], which contains a variety of generators that can be combined to create a wide range of SoC configurations.

We compile a large number of Chipyard SoC configurations to Verilog, aiming to collect Verilog files generated from as many Chisel source files as possible in the Chipyard repository. Our dataset includes 55% of the .scala files in Chipyard and its subrepositories; of those not covered, most do not contain synthesizable Chisel.

The generated Verilog contains annotations that indicate which Chisel file and line each Verilog line is generated from, allowing us to collect a set of relevant Chisel files for each generated Verilog file. Each Verilog file contains one module. We show an example of how Chisel-Verilog pairs are collected for one SoC configuration in Figure 2. Duplicates

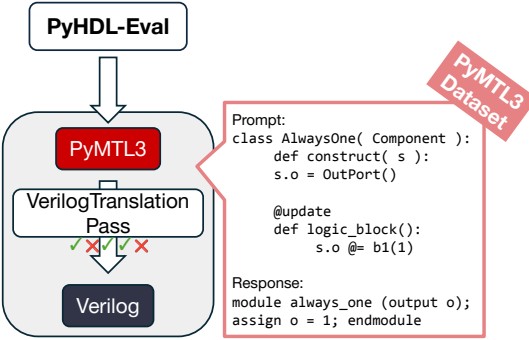

Fig. 3. How the PyMTL3 dataset is collected.

are removed, but for cases where the same Chisel file with different parameters generates differing Verilog output, all the data is kept. This results in 18,939 Chisel/Verilog pairs (with a total size of 1.69 GB). We additionally collect the corresponding FIRRTL. We do not use it for any further experiments in this work, but include it in our dataset for future users.

In our prompt/response pairs for LLM fine-tuning, the response consists of one generated Verilog file, which contain a single Verilog module. The prompt contains one or multiple Chisel source files (including one primary class and all of its dependencies), and a request to translate the code into Verilog. While the Chisel source code provided in the prompt does not directly compile to the response Verilog, we seek to improve the LLM's ability to correlate high-level HDL code in the Scala-embedded Chisel, which may contain more information about design semantics, to lower-level compiled Verilog.

*3) PyMTL3:* PyHDL-Eval [4] evaluates the ability of LLMs to correctly generate the Python-embedded HDLs PyMTL3, PyRTL, MyHDL, Migen, and Amaranth. The authors provide as an artifact the PyHDL code generated by LLMs during these experiments.

As shown in Figure 3, we use LLM-generated PyMTL3 code from PyHDL-Eval to construct a dataset similar to the Chisel dataset above. Like Chisel, PyMTL3 is a high-level HDL which can be compiled to Verilog. We compile each PyMTL3 example from PyHDL-Eval's artifact, numbering about 50,000, to Verilog using PyMTL3's VerilogTranslation-Pass. 18,636 examples (with a total size of 28 MB) compile to Verilog successfully, as many of the PyMTL3 examples in PyHDL-Eval contain syntax errors. In this dataset, each prompt is one PyMTL3 class, and each response is the corresponding translated Verilog module.

## IV. EXPERIMENT SETUP

### A. LLM Fine-Tuning Setup

We use Qwen2.5-Coder-32B-Instruct [17] as our base model. To train the model, we use DeepSpeed-Chat's Supervised Fine-Tuning pipeline [34] and enable ZeRO Stage 3 [30] and LoRA [16] for efficient training. We maintain consistent hyperparameter settings across all experiments, including the

use of FusedAdam optimizer, cosine learning decay, a learning rate of 1e-5, a single training epoch, and a batch size of 8. All experiments are conducted on a server with four NVIDIA L40S GPUs. A summary of the training hyperparameters is provided in the following table:

| Hyperparameter | Value |
|---|---|
| ZeRO Stage | 3 |
| LoRA Dimension | 32 |
| Data Type | bfloat16 |
| Batch Size | 8 |
| Learning Rate | 1e-5 |
| Number of Epochs | 1 |

TABLE I
HYPERPARAMETERS USED FOR FINE-TUNING

## B. Evaluation Setup

VerilogEvalV2 [29] is a benchmark that consists of 156 Verilog design problems, intended to test LLMs' ability to generate functionally correct Verilog according to a (mostly) natural language specification. We use VerilogEvalV2's `spec-to-rtl` benchmark to evaluate our model, with the following settings:

| Parameter | Value |
|---|---|
| Samples | 20 |
| Temperature | 0.85 |
| top_p | 0.95 |
| ICL examples | 0 |
| ICL rules | no |

TABLE II
PARAMETERS USED FOR EVALUATION WITH VERILOGEVALV2

We use both pass@1 and pass@10 to evaluate model performance. pass@1 effectively measures the total percentage of functionally correct responses, whereas pass@10 estimates the model's ability to generate at least one correct response when multiple samples (in this case 10) are taken.

## V. FINE-TUNING EXPERIMENTS

### A. Fine-Tuning with Individual Datasets

As shown in Figure 4, our datasets have varying effectiveness in improving our models' performance in VerilogEvalV2. Of the five datasets tested, PyMTL3 and VHDL perform the best, both providing about 18% increase in pass@10 over the base Qwen2.5-Coder-32B-Instruct model.

As we will discuss further in Section VI, the VHDL dataset has the highest perplexity in the dataset and provides a diverse dataset that has not been seen (as Verilog) during pretraining. On the other hand, while the PyMTL3 dataset is relatively less diverse, the set of designs it targets is highly relevant to VerilogEval, as PyHDL-Eval also generated code for a benchmark set of designs similar to VerilogEval. The entries in the PyMTL3 dataset also tend to be shorter than the entries in the other two hdl2v datasets (see Table III), making it easier for the model to learn relationships between the high-level PyMTL3 and the translated Verilog.

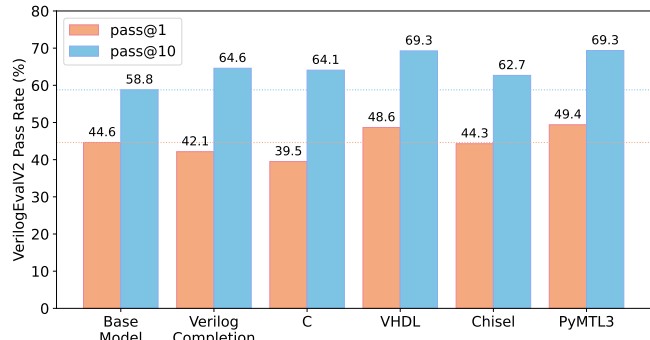

Fig. 4. VerilogEvalV2 performance for Qwen2.5-Coder-32B-Instruct, after being fine-tuned with each individual dataset.

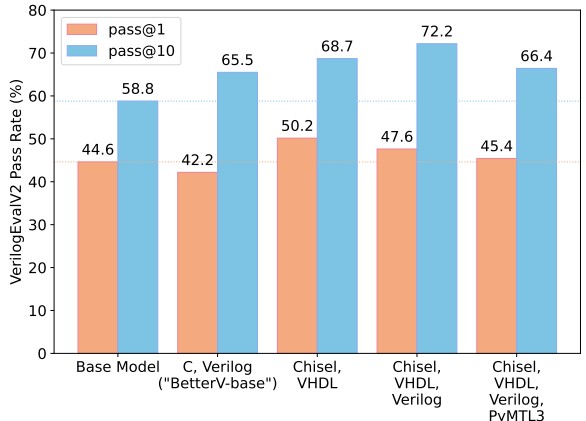

Fig. 5. VerilogEvalV2 performance for Qwen2.5-Coder-32B-Instruct, after being fine-tuned with combined datasets.

The Verilog, C, and Chisel datasets provide relatively smaller improvements. In fact, the C and Verilog Completion datasets decrease pass@1, but increase pass@10, which indicates that fine-tuning on these datasets has increased the diversity of generated code.

### B. Combining Datasets

We also explore the effects of fine-tuning with multiple datasets combined.

First, we fine-tune with our equivalent of BetterV's fine-tuning dataset. This includes the C dataset with both directions (C in the prompt and Verilog in the response, and vice versa), as well as our Verilog Completion dataset. Note that this data is not exactly the same as the dataset used in BetterV, and we do not include the discriminative guidance component.

Combining the C and Verilog datasets did not yield a significant improvement (within about one percentage point) compared to just Verilog. This is likely because these datasets originate from the same Verilog data, and as we show in Section VI-B, there are other languages that might perform better than C when translated to.

Next, we fine-tune data with other combinations of datasets. Specifically, we interleave datasets one entry at a time such that the distribution in the beginning of fine-tuning (when

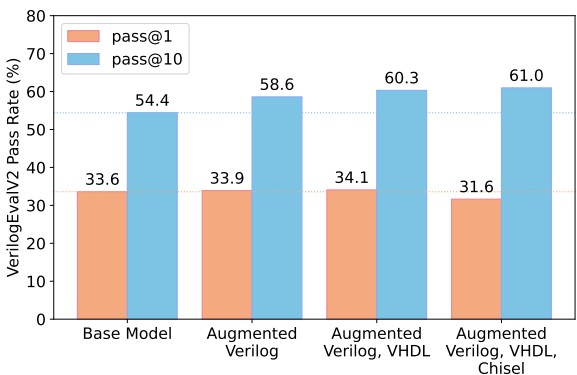

Fig. 6. VerilogEvalV2 performance of Qwen2.5-Coder-7B-Instruct, after being fine-tuned with gpt-4o-augmented versions of our datasets.

learning rate is highest) is equal for each dataset used. Combining datasets seems to yield limited but positive results. In particular combining Chisel and VHDL datasets yields our highest pass@1 of 50.2%, and combining Chisel, VHDL, and Verilog yields our highest pass@10 of 72.2%, as shown in Figure 5. However, adding PyMTL3 on top of this combination reduces both pass@1 and pass@10.

Overall, we find that fine-tuning with hdl2v data tends to increase both pass@1 and pass@10, but especially increases pass@10 (by 23%, compared to up to 13% increase in pass@1). Since hdl2v data originates from a variety of non-Verilog sources, it makes sense that fine-tuning with this data tends to improve the diversity of generated code, thereby increasing the likelihood of sampling at least one correct response among multiple.

### C. Data Augmentation

We further explore the potential benefits of hdl2v data via a data augmentation case study. Several prior works have shown that augmenting existing Verilog data using methods such as LLM-based summarization can be used to improve LLM Verilog generation performance via fine-tuning [7], [28], [33], [35].

In this work, we apply a generic data augmentation approach as a case study to demonstrate the usefulness of hdl2v data. Specifically, we prompt OpenAI's gpt-4o to generate natural language descriptions of Verilog modules, using OriGen's [10] code description prompt. We apply this prompt to the Verilog modules in the Verilog dataset, as well as the translated Verilog modules in the VHDL and Chisel datasets. We compare the effectiveness of fine-tuning with description-Verilog pairs from just the Verilog dataset to the effectiveness of fine-tuning with description-Verilog pairs from both the Verilog dataset and hdl2v datasets. Note that from this section onward, we use Qwen2.5-Coder-7B-Instruct (the 7B variant rather than 32B) as our base model, due to the high computational and runtime costs of fine-tuning.

We begin by fine-tuning with the augmented Verilog data, which yields a few percentage points improvement in pass@10 from 54.4% to 58.6%, as shown in Figure 6. Then, we add

gpt-4o-augmented Verilog data from hdl2v's VHDL dataset, which further boosts pass@10 from 58.6% to 60.3%. Combining augmented Verilog from the Verilog, VHDL, and Chisel datasets (the combination which yielded the highest pass@10 in Section V-B) boosts pass@10 even further to 61%, but drops pass@1 to 31.6%. Compared to the baseline of 54.8%, including hdl2v data increases the delta caused by fine-tuning by up to 63%, from 3.8 percentage points to 6.6 percentage points. This case study points to hdl2v being useful not just in isolation, but also in tandem with other approaches.

## VI. ANALYSIS

Our datasets differ along several axes. In addition to language, they also vary in factors such as distribution of designs and human readability. In this section, we characterize our datasets and perform two ablation studies to better understand what makes a dataset helpful in improving Verilog generation.

### A. Dataset Statistics

We characterize the **Verilog** code of each dataset to eliminate effects of language syntax. We use Qwen2Tokenizer to compute token counts and diversity. Table III includes statistics such as:

- **Type-Token Ratio (TTR)**, the ratio of unique tokens to total tokens.
- **N-gram diversity**, the ratio of unique token sequences of length N to the total number of such sequences. Higher values indicate greater token variety for both metrics.
- **Perplexity**, which measures how well a model makes predictions on a dataset, with lower values indicating better performance. The model's prediction accuracy can be estimated using the formula: $\frac{1}{\text{perplexity}} \times 100$. For example, a perplexity of 1.81 corresponds to a prediction accuracy of about 55.2%. In our case, we use Qwen2.5-Coder-7B as our model and randomly sample 1000 entries from each dataset to compute perplexity.

The Verilog code from the VHDL dataset has the highest perplexity of any of our datasets, and the highest vocabulary size and token diversity of the hdl2v datasets. This makes sense as the Verilog from the VHDL dataset is both unseen in pre-training data and is sourced from a wide range of repositories on GitHub. As a result, it is unsurprising that the VHDL dataset is one of the better-performing individual datasets (along with the PyMTL3 dataset) in Section V-A.

### B. C vs VHDL

In Section III-A2, we created our C-Verilog fine-tuning dataset by translating Verilog to C. In order to isolate the effects of using different languages from other variables, we create a dataset of Verilog to VHDL translations using this same dataset. We translate Verilog to VHDL using Icarus Verilog [32]. Both C and VHDL datasets are machine-translated, and they sample the same distribution of designs.

We train models using the subset of 12,612 pairs that were able to be translated to both C and VHDL. Like in Section V-C, we use Qwen2.5-Coder-7B-Instruct as our base

| Metric | | Verilog Completion | C | VHDL | Chisel | PyMTL3 |
|---|---|---|---|---|---|---|
| Total Tokens | | 27,818,433 | 5,274,385 | 7,039,588 | 128,662,957 | 6,607,407 |
| Vocabulary Size | | 23,247 | 15,075 | 22,279 | 4,441 | 1,731 |
| Type-Token Ratio (TTR) | | 0.0008 | 0.0029 | 0.0032 | 0.0000 | 0.0003 |
| N-gram Diversity | 2-gram | 0.0195 | 0.0393 | 0.0407 | 0.0003 | 0.0017 |
| | 3-gram | 0.0767 | 0.1217 | 0.1032 | 0.0010 | 0.0046 |
| Average Entry Length (no. tokens) | | 188.06 | 195.78 | 783.71 | 7394.64 | 353.55 |
| Standard Deviation of Entry Length | | 279.89 | 438.75 | 1796.34 | 10804.84 | 284.06 |
| Perplexity | | 1.81 | 2.15 | 2.34 | 1.55 | 1.84 |

TABLE III
STATISTICS FOR INDIVIDUAL DATASETS

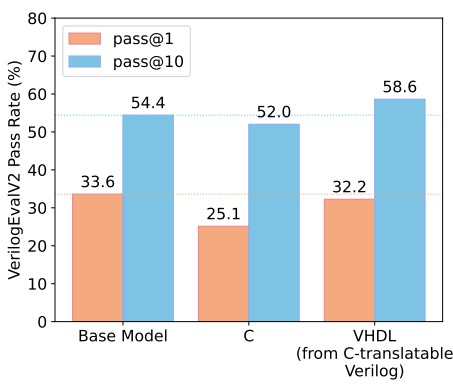

Fig. 7. VerilogEvalV2 performance for Qwen2.5-Coder-7B-Instruct, after being fine-tuned with a subset of the Verilog dataset translated to C and VHDL, respectively.

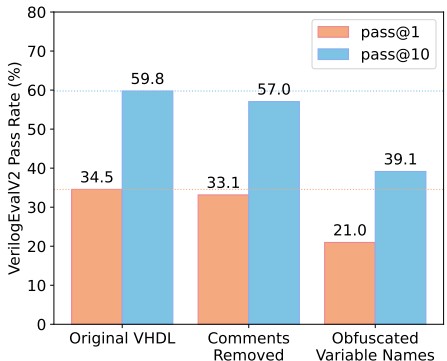

Fig. 8. VerilogEvalV2 performance for Qwen2.5-Coder-7B-Instruct, after being fine-tuned with modified versions of our VHDL dataset.

model. Figure 7 shows that the VHDL-translated dataset performs noticeably better than the C dataset. This indicates that the syntactic closeness of VHDL to Verilog, and the fact that VHDL is an HDL (as opposed to C, which is a software language) plays some role in our VHDL dataset outperforming our C dataset.

### C. Modifying the VHDL Dataset

In this section, we explore another axis of difference between datasets: human-readability. As a first step, we remove any comments from the VHDL dataset, then fine-tune Qwen2.5-Coder-7B-Instruct with this modified dataset. Next, in addition to removing comments, we obfuscate variable names across both VHDL and Verilog by replacing variable

names with generic placeholders. VHDL/Verilog keywords are preserved and common variable names across a VHDL-Verilog pair remain identical, but obfuscated.

As shown in Figure 8, we find that removing comments has only a small effect, whereas obfuscating variable names significantly degrades model performance. This shows that our model learns mostly from code, and the effect of natural language descriptions in the code (in the form of comments) is minimal.

## VII. CONCLUSION

In this work, we present hdl2v, which contains three new datasets for LLM Verilog generation fine-tuning. We utilize existing VHDL, Chisel, and PyMTL3 code to construct these datasets, and show that fine-tuning on HDL-Verilog translation pairs yields up to 13% improvement in pass@1 and 23% improvement in pass@10 on VerilogEvalV2. Furthermore, we find that some languages are inherently better than others for this process; specifically, we find that VHDL-Verilog pairs perform better than C-Verilog pairs for the same set of designs. We also find that the model does indeed learn from code rather than from the natural language comments in the code.

While hdl2v succeeds in improving LLM Verilog generation via fine-tuning, its real strength is the novel Verilog data it provides. Unlike prior work which focuses on augmentation of existing Verilog, we create multiple datasets of entirely new Verilog, which are both unseen in LLM pre-training corpora and are not generated by LLMs themselves. We demonstrate the value of this approach by combining hdl2v VHDL and Chisel data with the existing Verilog corpus to boost the performance of data augmentation-based fine-tuning by 63%. In future work, we would like to combine our dataset with other data augmentation methods, reasoning models, and agentic flows to push Verilog generation performance of open-weight models to even higher levels.

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
