# OpenReview forum: "hdl2v: A Code Translation Dataset for Enhanced LLM Verilog Generation"
_iscaconf.org/ISCA/2025/Workshop/MLArchSys — MLArchSys 2025 Oral_

### Official Review · Reviewer_PpiV · 2025-05-12
**This paper proposes hdl2v, a dataset designed to improve Verilog code generation by LLMs.**

**Confidence:** 3
**Rating:** 5

**Detailed Feedback And Questions For Authors:**

This paper addresses a critical bottleneck in LLM-based Verilog generation by introducing hdl2v, a dataset comprising translated Verilog examples from VHDL, Chisel, and PyMTL3. The work is timely, and impactful. The dataset construction is described in a reproducible manner, and the release of open-source resources adds value to the community. However, some aspects of the methodology and analysis could be strengthened to improve clarity.

First, while the dataset sizes are clearly reported, there is little discussion of design diversity, such as coverage across hardware domains, complexity levels, or coding styles. Many examples come from constrained sources (e.g., Chisel via Chipyard), which may bias the dataset toward a narrow subset of hardware structures. This makes it difficult to assess how well fine-tuning with hdl2v generalizes to unseen real-world Verilog. Also, Table III presents metrics such as token diversity and perplexity, but the paper does not draw meaningful insights from them. In fact, the authors state no correlation was observed between these statistics and model performance.

My other comment is about the interpretability of VerilogEvalV2 pass rates. While the pass@1 and pass@10 improvements are compelling, the paper assumes familiarity with VerilogEvalV2 without explaining what constitutes a "pass." This might affect the reader’s ability to evaluate whether the metric truly captures correctness, synthesisability, or functionality. A brief summary or example of how VerilogEvalV2 scoring works would make the results more self-contained.

Finally, although translation tools (e.g., vhd2vl, PyMTL3 VerilogTranslationPass) are mentioned, there is no discussion of translation error rates or manual validation. Since many datasets are machine-generated, the paper should probably clarify whether any quality control was applied to ensure semantic equivalence between source HDL and generated Verilog.

**Top Reasons To Accept The Paper:**

1. Introduces a valuable open-source dataset addressing Verilog scarcity in LLM training.

2. Offers a reproducible methodology and thoughtful dataset engineering pipeline.

3. Provides initial analysis of which source HDLs contribute most effectively to Verilog generation performance.

**Top Reasons To Reject The Paper:**

1. Limited justification of dataset coverage and generality; lacks evidence of representativeness.

2. Statistical analysis is superficial and does not drive strong insights.

3. Quality control of translated datasets is unclear.

---

### Official Review · Reviewer_auyj · 2025-05-16
**hdl2v Review**

**Confidence:** 4
**Rating:** 7

**Detailed Feedback And Questions For Authors:**

This paper presents hdl2v, a dataset to augment Verilog inputs for LLM training to help LLMs better generate Verilog code. The authors clearly put a lot of work into this effort, and I think this is a valuable contribution that will help future research in this area.

Notes / Questions:

- Is translation always necessary? For example, I am wondering if you could directly feed in VHDL examples  (instead of translating the VHDL to Verilog) and ask the LLM (through prompting) to pay attention to code structure and the logic but not necessarily specific syntax, and if that would still helpful generate functional Verilog code at the end. Since both VHDL and Verilog are hardware description languages, I think the LLM may be able to grasp hardware concepts and hardware module structures at a higher level from the VHDL directly.

- The authors mention that "utilizing meaningful module and variable names is important in helping LLMs learn from this data". In that thread, I am curious how helpful high-quality comments in the code are. I see that the authors do an ablation study in removing comments, and it is interesting to see that there is only a small impact without comments, but I am curious about the effect of the quality / length of the comments. It may be interesting to do a further ablation study looking at code to comment ratios and seeing which ratios do (or do not) impact performance.

- To further augment the dataset, I am curious if it would be helpful to feed in "bad" examples / failed examples of Verilog for the model to learn what not to do as well. For example, if it generates Verilog that is not functional, it could be useful to feed that back into the model during fine-tuning as a bad example instead of only strictly feeding in good examples.

- As mentioned earlier: While the authors use quantitative metrics to evaluate their dataset and subsequent training, the paper does not consider a qualitative analysis of the Verilog generated by the fine-tuned model. Is the Verilog generally human-readable (which can sometimes be an issue with Python-based/other-language-based Verilog generators and translation tools)? I realize that this "quality" is difficult to measure, but I think a qualitative discussion would be interesting.

- As mentioned earlier: There is also not discussion on whether the generated Verilog will be synthesizable and be functional in the downstream physical design flow. This will be an important metric to evaluate the generated Verilog.

**Top Reasons To Accept The Paper:**

- The core contribution of the paper, the hdl2v dataset, is valuable in augmenting the Verilog data available to train LLMs. The authors are open-sourcing this dataset, which will likely be helpful for future research.
- With their dataset with Qwen2.5-Coder-32B-Instruct, the authors get improved performance by up to 23% (pass@10) in VerilogEvalV2, demonstrating the utility of their dataset.
- The paper is generally well-written, and the ablation studies at the end add further insight into how to augment this dataset further.

**Top Reasons To Reject The Paper:**

- While the authors use quantitative metrics to evaluate their dataset and subsequent training, the paper does not consider a qualitative analysis of the Verilog generated by the fine-tuned model. Is the Verilog generally human-readable (which can sometimes be an issue with Python-based/other-language-based Verilog generators and translation tools)? I realize that this "quality" is difficult to measure, but I think a qualitative discussion would be interesting.
- There is also not discussion on whether the generated Verilog will be synthesizable and be functional in the downstream physical design flow.

---

### Official Review · Reviewer_TMSQ · 2025-05-18
**Better data for HDL to Verilog Generation**

**Confidence:** 5
**Rating:** 5

**Detailed Feedback And Questions For Authors:**

This is a useful application of code translation, and the domain is important. The insight into using "nearby" languages to augment the language of choice (Verilog) is very well received by this reviewer. However, it would've been better to explore more than one model.

Another high level question: how do you know that the dataset has "good" code? Syntactic correctness is one thing (which, historically, shouldn't be that big of an issue for language models after seeing enough samples); but how about functional correctness of the dataset? Is there a way to ensure it, or all the dataset has to be executed in Verilog for checking? Is there some notion of a test bench or unit tests that can be employed to measure the goodness of the dataset?

**Top Reasons To Accept The Paper:**

This is an important problem, as Verilog code is quite important but currently does not have as much data as popular high level languages (such as C or Python). The authors insight to align it to something nearby, namely VHDL, can substantially help; and that is what the paper's approach and results are centered around. Results are Ok.

**Top Reasons To Reject The Paper:**

Only one model family is explored, Qwen2.5-coder. This makes it difficult to fully understand whether the improved results come from the better data collection, or Qwen2.5coder's overall impressive abilities. Good data is certainly important, but how much relative to choosing the right/best model? A simple ablation study or inclusion of other models with and without the new dataset would help empirically understand this difference.